# Gender- and Grade-Dependent Activation of Androgen Receptor Signaling in Adult-Type Diffuse Gliomas: Epigenetic Insights from a Retrospective Cohort Study

**DOI:** 10.3390/biomedicines13102379

**Published:** 2025-09-28

**Authors:** Lidia Gatto, Sofia Asioli, Luca Morandi, Enrico Di Oto, Vincenzo Di Nunno, Alicia Tosoni, Marta Aprile, Stefania Bartolini, Lucia Griva, Sofia Melotti, Francesca Gentilini, Giuseppe Pinto, Francesco Casadei, Maria Pia Foschini, Caterina Tonon, Raffaele Lodi, Enrico Franceschi

**Affiliations:** 1Nervous System Medical Oncology Department, IRCCS Istituto delle Scienze Neurologiche di Bologna, Bellaria Hospital, 40139 Bologna, Italy; vincenzo.dinunno@isnb.it (V.D.N.); a.tosoni@isnb.it (A.T.); marta.aprile@ausl.bologna.it (M.A.); stefania.bartolini@isnb.it (S.B.); e.franceschi@isnb.it (E.F.); 2Department of Biomedical and Neuromotor Sciences (DIBINEM), University of Bologna, 40136 Bologna, Italy; sofia.asioli3@unibo.it (S.A.); luca.morandi2@unibo.it (L.M.); lucia.griva@studio.unibo.it (L.G.); francesca.gentilini4@studio.unibo.it (F.G.); giuseppe.pinto4@studio.unibo.it (G.P.); francesco.casadei@aosp.bo.it (F.C.); caterina.tonon@unibo.it (C.T.); raffaele.lodi@unibo.it (R.L.); 3IRCCS Istituto delle Scienze Neurologiche di Bologna, 40139 Bologna, Italy; 4Functional and Molecular Neuroimaging Unit, IRCCS Istituto delle Scienze Neurologiche di Bologna, Bellaria Hospital, via Altura 3, 40139 Bologna, Italy; 5Solid Tumor Molecular Pathology Laboratory, IRCCS Azienda Ospedaliero-Universitaria di Bologna, 40138 Bologna, Italy; 6Department of Medical and Surgical Sciences (DIMEC), University of Bologna, 40138 Bologna, Italy; 7Pathology Unit, IRCCS Azienda Ospedaliero-Universitaria di Bologna, 40138 Bologna, Italy; sofia.melotti@ausl.bologna.it; 8School of Medicine, Dipartimento di Scienze Mediche e Chirurgiche (DIMEC), University of Bologna, 40138 Bologna, Italy; 9Department of Biomedical and Neuromotor Sciences, Section of Anatomic Pathology at Bellaria Hospital, University of Bologna, 40138 Bologna, Italy; mariapia.foschini@unibo.it; 10Functional and Molecular Neuroimaging Unit, IRCCS Istituto delle Scienze Neurologiche di Bologna, 40139 Bologna, Italy

**Keywords:** androgen receptor, DNA methylation, enzalutamide, glioblastoma, glioma

## Abstract

**Background**: The androgen receptor (AR) is a ligand-dependent transcription factor of the nuclear steroid receptor superfamily, implicated in the pathogenesis of various solid tumors. The *AR* gene, located on chromosome Xq11–12, is accompanied by several X-linked genes that modulate *AR* expression and function, including *FLNA*, *UXT*, and members of the melanoma antigen gene (MAGE) family (*MAGEA1*, *MAGEA11*, *MAGEC1*, *MAGEC2*). While the AR has been investigated in multiple tumor types, its role in adult-type diffuse gliomas remains largely unexplored. Here, we characterized AR protein expression and the promoter methylation status of the AR and associated regulatory genes in adult-type diffuse gliomas. **Methods**: A retrospective analysis was conducted on 50 patients with adult-type diffuse gliomas, including IDH-mutant gliomas (grades 2–4) and IDH-wildtype glioblastomas (GBMs), classified according to the 2021 WHO criteria. AR nuclear expression was assessed by immunohistochemistry (IHC). Methylation-specific PCR and quantitative DNA methylation analyses were employed to evaluate promoter methylation of the AR and selected co-regulatory genes. **Results**: AR nuclear positivity correlated significantly with male sex (*p* = 0.04) and higher tumor grade, with the highest expression in IDH-wildtype GBMs (*p* = 0.04). In IDH-mutant gliomas, AR immunoreactivity was more prevalent in astrocytomas than in 1p/19q codeleted oligodendrogliomas (*p* = 0.02). AR expression was associated with unmethylated MGMT promoter status (*p* = 0.02). DNA methylation analysis revealed AR gene hypomethylation in tumors displaying nuclear AR positivity and in IDH-wildtype GBMs (Kruskal–Wallis *p* < 0.05). Additionally, methylation patterns of AR co-regulators located on the X chromosome suggest epigenetic regulation of AR signaling in gliomas. **Conclusions**: The findings reveal distinct AR pathway activation patterns in adult-type diffuse gliomas, particularly IDH-wildtype GBMs, suggesting that further exploration of antiandrogen therapies is warranted.

## 1. Introduction

Adult-type diffuse gliomas are the most frequent tumors of the central nervous system (CNS), characterized by an extensive infiltrative growth with an overall survival that greatly varies depending on the specific subtype and histological grade, ranging from 7 to 15 years in Isocitrate Dehydrogenase (*IDH*)-mutant low-grade gliomas (LGGs) to 12–24 months in grade 4 *IDH*-wildtype gliomas [1,2]. Despite surgery, radiotherapy, and alkylating agents being the cornerstones of glioma treatment, effective treatments are lacking due to challenges such as the inability to fully resect infiltrative tumors, treatment resistance, difficulties delivering drugs across the blood–brain barrier (BBB), and tumor heterogeneity.

Although existing studies have tried different therapeutic methods [3], individually tailored strategies are needed to improve outcomes in subsets of patients [4,5,6,7].

A significant gender disparity exists in the incidence of glioblastoma (GBM), with a higher prevalence in males compared with females. Research suggests that this difference may be linked to the modulation of glioblastoma pathogenesis via the androgen receptor (AR) signaling pathway. Dysregulated activation of the AR signaling cascade, or cross-talk between the AR and other oncogenic signaling networks, appears to facilitate GBM progression. Consequently, AR inhibition or modulation presents a potential therapeutic strategy for GBM management [8].

Androgens interact with the AR to produce their biological effects, and they are essential for gender differentiation and male phenotypic development [9,10].

When androgens, primarily testosterone and dihydrotestosterone, bind to the AR, it becomes active. The inactivated AR is mainly located in the cytoplasm; the binding of the AR to androgens produces a conformational change in the receptor that dimerizes, migrates into the nucleus, and binds to specific hormone response elements to regulate the expression of androgen-responsive genes [11].

Several studies have shown that the AR is overexpressed in some human malignancies, such as prostate cancer, breast cancer, hepatocellular carcinoma, and bladder cancer [9,12,13]. However, to date, the same relationship is not as well understood in gliomas. Recent research suggests a role for the AR in gliomagenesis [14]; however, the expression level of the AR, its regulatory mechanisms, and its prognostic significance in gliomas remain unknown [15].

The *AR* gene is situated on the X chromosome at the Xq11–12 locus [16] and *AR* gene expression is modulated by regulators, predominantly located on the X chromosome, including filamin A (*FLNA*), Ubiquitously Expressed Prefoldin-Like Chaperone (*UXT*) and the genes belonging to melanoma antigen (*MAGE*) family genes, *MAGEA1*, *MAGEA2*, *MAGEA3*, *MAGEA9*, *MAGEA11*, *MAGEC1*, and *MAGEC2* [17,18,19].

DNA methylation is a crucial epigenetic regulatory mechanism for gene transcription and represents a significant area of investigation in oncology. It is linked to aberrant gene silencing by sustaining a transcriptionally repressed state characterized by heterochromatin formation [20]. Methylation involves cytosine on the CpG sequences, which are abundant in the promoter regions of genes. The result is 5-methylcytosine, which produces transcriptional repression and silences gene expression [21].

In addition, the *AR* gene and its regulators, which map to the X chromosome, may exhibit copy number variations as a consequence of X-chromosome aneuploidy. At present, there is no available data regarding the variations in copy number of the X chromosome and AR in gliomas.

This study aimed to evaluate the AR nuclear expression detected at immunohistochemistry and to correlate AR expression with the data emerging from DNA sequencing, DNA methylation analysis, and X-chromosome copy number analysis. In particular, we assessed the methylation pattern of the *AR* gene and its regulatory genes such as *FLNA*, *UXT*, and *MAGE* family members, and analyzed X-chromosome copy number variations in a series of gliomas.

## 2. Materials and Methods

### 2.1. Study Design

This is a retrospective study aimed at assessing AR immunohistochemical expression, AR methylation patterns, and X-chromosome AR-related gene methylation patterns in adult-type diffuse gliomas.

We analyzed 50 tissue samples obtained from patients who underwent surgery at Bellaria Hospital in Bologna, between 1999 and 2021. All specimens were consistently preserved at the pathology department of our institution.

All cases were independently re-evaluated, classified, and graded by a neuropathologist (A.S.) according to currently available criteria (CNS WHO 2021).

Inclusion criteria were as follows:(1)age ≥ 18 years old;(2)morphological and immunohistochemical profile consistent with the diagnosis of adult-type diffuse glioma (*IDH*-mutant glioma, grade 2 or 3 or 4 and GBM *IDH*-wildtype grade 4 according to the fifth edition of CNS WHO 2021 [2]);(3)histological slides/formalin-fixed paraffin-embedded tissue tumor (FFPE) blocks from the archive available to perform immunohistochemical assessment of androgen receptor, in situ hybridization, and molecular tests.

Exclusion criteria were as follows:(1)histological diagnosis different from adult-type diffuse gliomas;(2)unavailability of histological slides/formalin-fixed paraffin-embedded tissue tumor (FFPE) blocks;(3)molecular GBM *IDH*-wildtype and gliomas *IDH*-mutant with homozygous cyclin-dependent kinase inhibitor (CDKN2A/B) gene deletion were excluded from our study;(4)patients undergoing antiandrogen therapy for another malignancy.

The primary objective of the study was to assess nuclear AR expression by immunohistochemistry on tissue samples, and how AR expression might be influenced by the DNA methylation pattern of the *AR* and its co-regulators and by X-chromosome copy number variations.

The secondary purpose of the study was to explore the prognostic value of the AR in adult-type diffuse gliomas.

The impact of categorical variables on time-dependent events such as overall survival (OS) and progression-free survival (PFS) has been estimated through the Kaplan–Meier method [22].

### 2.2. Ethical Statement

The study was approved by the Ethics Committee of the Azienda Sanitaria Locale di Bologna (protocol number CE09113, Bologna, Italy). This study was conducted in agreement with the most recently updated Declaration of Helsinki and all the international and local laws applied to clinical trials and patient protection. The study was conducted according to the principles of the ICH Harmonized Tripartite Guideline for Good Clinical Practice.

### 2.3. AR Immunohistochemistry

AR nuclear expression was evaluated in tissue sections using immunohistochemistry. Samples were fixed in 10% neutral-buffered formalin for 24 h and processed into formalin-fixed paraffin-embedded (FFPE) blocks following standard protocols. Immunostaining was conducted on an automated platform (Ventana BenchMark, Ventana Medical Systems Inc., Tucson, AZ, USA) utilizing a pre-diluted monoclonal anti-AR antibody (Cell Marque, Rocklin, CA, USA, clone SP107). Only nuclear immunoreactivity was considered for scoring. AR expression was assessed using a semi-quantitative four-point scoring system adapted from astrocytic tumor analyses [23], ranging from 0 to 3. Score 3 indicated strong, widespread (>50%) nuclear staining; score 2 denoted focal nuclear staining in approximately 50% of cells; score 1 was assigned for weak or focal nuclear staining in <50% of cells or faint staining not readily discernible at low magnification; and score 0 indicated absence of staining [23]. The highest AR staining intensity observed in adjacent non-neoplastic glial tissue served as the internal reference. In cases with heterogeneous staining, the predominant score was recorded. Scoring was performed independently by a neuropathologist (AS) blinded to clinical data. Tumors were classified as AR-positive (scores 1–3) or AR-negative (score 0).

### 2.4. Next-Generation Sequencing

Genomic DNA was extracted from formalin-fixed, paraffin-embedded (FFPE) tumor sections using the NucleoSpin^®^ Tissue Kit (Macherey-Nagel, Dueren, Germany), following the manufacturer’s protocol. A board-certified neuropathologist (AS) reviewed the histological slides and identified the most representative tumor regions for analysis. Targeted mutational profiling was performed via next-generation sequencing (NGS), focusing on the IDH1, IDH2, TERT, and AR genes [24]. FASTQ data were filtered based on PHRED quality scores (>Q30) and processed using the Galaxy Project platform [25], incorporating alignment to the hg38 human reference genome with Bowtie2, local realignment with GATK, variant calling via HaplotypeCaller, and duplicate read removal using Picard MarkDuplicates. BAM files were visualized with the Integrative Genomics Viewer (IGV). Variants with a variant allele frequency (VAF) > 20% were retained. Functional predictions were assessed using PolyPhen-2 and cross-referenced with the COSMIC database [26]. All NGS assays were designed to achieve a minimum depth of coverage of 1000× with ≥1000 reads per region.

For DNA methylation analysis, bisulfite-converted NGS was employed to assess CpG methylation across multiple X-linked genes including the AR, MAGEA1, MAGEA11, MAGEC1, MAGEC2, UXT, and FLNA. Genomic DNA (50–500 ng) was treated with bisulfite using the EZ-96 DNA Methylation MagPrep Kit (Zymo Research, Irvine, CA, USA; Cat. D5040), per the manufacturer’s instructions. Specific CpG regions, including six within the AR gene (spanning from the first non-coding region to exon 1), were amplified using primers designed via MethPrimer (http://www.urogene.org/methprimer/, accessed on 24 September 2025). Libraries were generated using tagged primers and Phusion U DNA Polymerase (Thermo Fisher Scientific, Waltham, MA, USA, Cat. F555L), followed by a two-step PCR enrichment protocol utilizing the Nextera™ Index Kit (Illumina, San Diego, CA, USA) [27]. Sequencing was carried out on the MiSeq platform (Illumina), according to standard protocols. Methylation quantification was performed using BWAmeth for alignment and MethylDackel for methylation calling within the Galaxy environment. The methylation panel also included the TERT promoter and three specific regions of the MGMT gene.

Additionally, AR mutational analysis was extended to exons 4, 5, and 8—regions known to harbor frequent mutations associated with therapeutic resistance in prostate cancer.

### 2.5. FISH Analysis

FISH analysis was performed to assess the level of X-chromosome aneuploidy of the glioma samples.

A neuropathologist (AS) examined the tissue sections and selected the most representative tumor area on the histological slides.

Dual-color FISH was carried out according to a standard protocol summarized as follows: five-micron sections were obtained from each tumor block. One specific probe kit for the X chromosome (AR gene amplification probe, (AR(Xq12)/CEPX OACP IE LTD, Cork, Ireland), added with the Smart-ISH hybridization buffer (OACP IE LTD, Cork, Ireland), was applied [17,28]. The AR gene-specific probe length is stated as 281 Kb (AR gene amplification probe product datasheet OACP IE LTD, Cork, Ireland).

## 3. Results

Fifty patients were included, 26 (52%) were male and 24 (48%) were female. The median age at diagnosis was 50 years (range 26–77 years).

Histology included astrocytoma *IDH*-mutant grade 3 (n = 3), oligodendroglioma *IDH*-mutant and 1p19q codeleted grade 3 (n = 4), astrocytoma *IDH*-mutant grade 2 (n = 9), oligodendroglioma *IDH*-mutant and 1p19q codeleted grade 2 (n = 11), and GBM *IDH*-wt grade 4 (n = 23).

AR positivity was detected in 21 out of 50 samples (42%). Among different glioma subtypes, AR positivity was found in fourteen patients diagnosed with GBM *IDH*-wildtype grade 4 (66.7%), in four cases with astrocytoma *IDH*-mutant grade 2 (19%), in two cases of astrocytoma *IDH*-mutant grade 3 (9.5%) and in one case of oligodendroglioma *IDH*-mutant and 1p19q codeleted grade 2 (4.8%) (Table 1 and Figure 1). AR positivity was higher in males (71.4%; n = 15) than in females (28.6%, n = 6).

The mutational analysis of the *AR* gene from exon 4 to 8 revealed no mutations in all samples.

The clinicopathological characteristics of the cases are summarized in Table 1.

### 3.1. Gender and Pathologic Grade Are Associated with AR Expression Levels in Adult-Type Diffuse Gliomas

In our analysis AR expression was correlated with gender and with glioma grade. Immunohistochemical AR positivity was predominant in males (*p* = 0.04) and in patients with GB *IDH*-wildtype compared with low-grade *IDH*-mutant adult-type diffuse gliomas (*p* = 0.04) (Figure 2A). Among *IDH*-mutant LGGs, AR expression was more frequent in *IDH*-mutant astrocytomas compared with *IDH*-mutant oligodendrogliomas and 1p19q codeleted (*p* = 0.02) (Figure 1A). Moreover, AR expression was more frequent in *MGMT*-unmethylated gliomas compared with *MGMT*-methylated tumors (*p* = 0.02) (Figure 2B).

### 3.2. Prognostic Role of AR Expression Levels in Adult-Type Diffuse Gliomas

Immunohistochemically AR-positive gliomas showed a lower overall survival (OS) than AR-negative gliomas (*p* = 0.0022; HR = 3.1 (95%CI 1.45–6.6)) (Figure 3).

Among the different histological entities, we found the following survival outcomes:

In *IDH*-wt AR-positive GBMs, the median OS was 12.3 months (10.6–37.7; *p*-value 0.22; HR = 1.74 (95%CI 0.71–4.45)). In *IDH*-wt AR-positive and MGMT-methylated GBMs, the mOS was 18 months (9.4-NA; HR = 1.73 (95%CI 0.5–6.02)), while in *IDH*-wt AR-positive MGMT-unmethylated GBMs patients the mOS was 11.5 months (10.3-NA; *p*-value 0.31; HR = 2.1 (95%CI 0.5–8-84)). In *IDH*-wt AR-negative GBMs, the median OS was 16.4 months (13.3-NA). In *IDH*-wt AR-negative and MGMT-methylated GBMs, the mOS was 30 months (7.4-NA), while in *IDH*-wt AR-negative MGMT-unmethylated GBMs, the mOS was 14.2 months (13.3-NA). In IDH-mutant AR-positive grade 3 astrocytomas, the mOS was 137 months (*p*-value 0.81; HR = 0.71 (95%CI 0.04–11.8)), while in *IDH*-mutant AR-negative grade 3 astrocytomas, the mOS was 128 months. In *IDH*-mutant AR-positive grade 3 oligodendrogliomas (n = 1), the OS was 144 months, while in AR-negative grade 3 oligodendrogliomas, the mOS was not assessable, as the events had not yet been reached. In grade 2 *IDH*-mutant astrocytomas and oligodendrogliomas, the mOS was not assessable, as the events had not yet been reached.

We performed a Cox Regression Model investigating the prognostic impact of AR expression adjusted for MGMT-methylation status in the GBM *IDH*-wt group (Table 2). In this analysis, the MGMT-methylation status confirmed its prognostic role (*p*-value 0.04; HR = 0.34 (0.12–0.95)), while AR expression did not reach statistical significance (*p*-value 0.27; HR = 1.68 (0.67–4.26)), likely due to the small sample size. After adjusting the Cox Regression Model for MGMT-methylation status and gender, AR expression only showed a trend in impacting survival, without reaching statistical significance (*p*-value 0.12; HR = 2.25 (0.79–6.39)) (Table 3).

### 3.3. Differential DNA Methylation Analysis of AR Gene Promoter Region According to Immunohistochemical AR Expression

In our series we found a relation between immunohistochemical AR expression and methylation levels of different CpG sites in the *AR* promoter.

The AR expression was analyzed in the samples through immunohistochemistry. The samples were divided into two groups: positive AR expression (group 0, AR expression score 1, 2, and 3) or negative AR expression (group 1, AR expression score 0), based on the H-score being higher than or equal to 0.

Comparative analysis of DNA methylation in gliomas with high vs. low immunohistochemical expression of the AR involved four different regions of the *AR* gene body: AR CAG repeats (promoter) from coordinates 67,545,204 to 67,545,299; the AR region within CAG and CGG repeats (intra CAG-CGG) from 67,545,904 to 67,546,136; the AR region before CGG repeat from 67,546,381 to 67,546,405; and, finally, the AR region within the CGG repeats from 67,546,492 to 67,546,664.

Appendix A illustrate the levels of methylation for each genetic region compared between the two groups, positive vs. negative immunohistochemical expression of the AR: positive AR expression (group 0) or negative AR expression (group 1).

The *AR* was found hypomethylated, and therefore probably transcriptionally active, in cases with immunohistochemical positivity of the AR (Kruskal–Wallis < 0.05). In males, *AR* CAG repeats and AR INTRA CAG-CGG repeats resulted hypomethylated in AR-positive gliomas. In females, we observed the same trend; however, it was not statistically significant.

### 3.4. Differential Methylation Analysis of MAGEA1, MAGEA11, MAGEC1, MAGEC2, UXT, and FNLA Genes According to Immunohistochemical AR Expression

In males, *MAGEA1* was found to be hypermethylated in cases negative for AR expression. In females, we observed a different trend: *MAGEA1* was found to be hypomethylated in cases with low AR expression (Appendix A).

*MAGEA11* showed a methylation profile similar to *MAGEA1*. In particular, in males *MAGEA11* was hypermethylated in cases negative for AR expression. On the contrary, in females *MAGEA11* was hypomethylated in cases with low AR expression (Appendix A).

The same trend was observed for the *MAGEC1* gene. *MAGEC1*, in males, showed a tendency to be hypermethylated in AR-negative cases. In females, on the contrary, *MAGEC1* showed a tendency to be hypomethylated in AR-negative cases (Appendix A).

MAGEC2 showed a tendency to be hypomethylated in cases negative for AR expression both in male and in female patients (Appendix A).

In males, UXT is hypermethylated in immune cases with positive AR (Kruskal–Wallis < 0.05). In females UXT shows the same trend, however it is not significant (Appendix A).

FLNA showed a tendency to be hypomethylated in AR-negative cases both in male and in female patients (Appendix A).

### 3.5. Differential Methylation Analysis of the MGMT Promoter According to Immunohistochemical Expression of AR

Regardless of gender, since *MGMT* is located on chromosome 10, we found that *MGMT* is hypermethylated in AR-positive cases (KR < 0.05) (Appendix A).

### 3.6. Differential Methylation Analysis of the AR Gene Promoter Region According to Pathologic Grade

In our analysis, AR INTRA CAG-CGG repeats and AR CAG repeats were found to be hypermethylated in low-grade gliomas and hypomethylated in high-grade tumors (Appendix A).

### 3.7. Differential Methylation Analysis of MAGEA1, MAGEA11, MAGEC1, MAGEC2, UXT, and FNLA According to Pathologic Grade

*MAGEA1* was found to be hypomethylated in low-grade tumors in male and in female patients (Appendix A).

*MAGEA 11* showed a different behavior based on gender. In particular, in males *MAGEA 11* was found to be hypermethylated in low-grade tumors; in females, on the contrary, it was found to be hypomethylated in low-grade tumors (Appendix A).

For *MAGEC 1*, in both males and females, we observed a trend towards hypermethylation in low-grade tumors (Appendix A).

Similarly, *MAGEC 2* and *FLNA* also resulted hypermethylated in low-grade tumors (Appendix A). Conversely, *UXT* was found to be hypomethylated in low-grade tumors and hypermethylated in high-grade tumors (Appendix A).

### 3.8. Differential Methylation Analysis of MGMT and TERT According to Pathologic Grade

*MGMT* showed higher methylation in lower-grade gliomas and a lower methylation in higher-grade gliomas in the enhancer region, while the opposite trend was found for the exon1 region. This suggests that the methylation pattern of *MGMT* may vary depending on tumor grading. Higher-grade tumors tend to be *MGMT* hypomethylated in the enhancer region and hypermethylated in the exon1 region (Appendix A).

Finally, *TERT* resulted hypomethylated and therefore actively expressed in high-grade gliomas, which is consistent with *TERT* overexpression in more aggressive tumors (Appendix A).

### 3.9. X-Chromosome Copy Number Analysis in Glioma

We performed a FISH analysis to assess the level of X-chromosome aneusomy. Our analysis showed that X-chromosome monosomy is more frequent in women (*p* < 0.001), while X-chromosome polysomy is more frequent in men (*p* < 0.001).

In our series polysomy resulted more frequently in GBM *IDH*-wt (*p* = 0.05) (Figure 4) and unmethylated tumors (*p* = 0.037) (Figure 5). In unmethylated patients there is also a trend of higher frequency of AR amplification (*p* = 0.06).

Table 1 shows the percentage of wildtype X chromosome, AR deletion, monoploidy of the X chromosome in female samples, amplification of the AR, and polyploidy of the X chromosome. Polyploidy means the presence of XX chromosomes in male samples and the presence of more than two X chromosomes in female samples.

The average polyploidy for females is 1.659%, while for males the average polyploidy is 15.745%.

### 3.10. Differential Methylation Analysis of AR Promoter Regions According to X-Chromosome Copy Number

We observed that a lower X-chromosome polyploidy was associated with intra CAG-CGG repeats hypermethylation (Appendix A).

### 3.11. Differential Methylation Analysis of MAGEA1, MAGEA11, MAGEC1, MAGEC2, UXT, and FNLA Genes According to X-Chromosome Copy Number

We observed a correlation between a lower X-chromosome polyploidy and the hypomethylation status of *MAGEA1*, *MAGEA11,* and *UXT* genes (Appendix A).

## 4. Discussion

The oncogenic role of the AR in classical hormone-responsive tissues such as prostate and male breast is well established [10,17,29,30]. The AR is a targetable molecule widely used in the treatment of prostate cancer [31,32,33,34,35,36]; however, its role in gliomas is still not well understood.

In recent years, more attention has been given to understanding the pathogenic role of AR signaling in glioma biology [37,38,39,40]. AR overexpression has been observed in gliomas and their stem-like cell compartments, supporting the idea that AR signaling may affect the development of brain tumors [15,41,42]. However, due to conflicting and inconclusive data in the literature, the prognostic significance of AR overexpression in gliomas remains uncertain.

In this study, we aimed to investigate AR expression levels across gliomas of different histopathologic grades and molecular subtypes (Table 1) and to analyze the DNA methylation level of the AR and its co-regulators mapped on the X chromosome.

In our series AR expression positively correlated with glioma grading and with gender. Our analysis reports a higher expression of the AR in males (71.4%; n = 15) compared with females (28.6%, n = 6), (*p* = 0.04) and in higher-grade tumors, in particular in GBMs *IDH*-wt compared with low-grade *IDH*-mutant diffuse gliomas (*p* = 0.04) (Figure 1). Among *IDH*-mutant LGGs, *AR* expression was more frequent in IDH-mutant astrocytomas compared with IDH-mutant oligodendrogliomas and 1p19q codeleted (*p*= 0.02) (Figure 2A). Furthermore, AR expression was more frequent in MGMT-unmethylated gliomas compared with MGMT-methylated tumors (*p* = 0.02) (Figure 2B).

This aligns with data from previous research indicating that AR activation influences the proliferation of high-grade glioma cells and, primarily, of GBM cells [15,43]. Several studies suggest that AR expression is positively correlated with tumor pathological grade, with grade 4 tumors demonstrating the most elevated expression levels [15,38,44,45]. In particular, the AR is detected in GBM *IDH*-wt samples compared to normal human brain tissue and is associated with a worse prognosis in GBM *IDH*-wt patients [15,46]. Data from The Cancer Genome Atlas (TCGA) database indicate an upregulation of the AR at both mRNA and protein levels in human GBM *IDH*-wt specimens, particularly among male patients, in comparison with normal brain tissue [47]. Epidemiology indicates a gender disparity in GBMs [8,48,49,50]: the Central Brain Tumor Registry of the United States (CBTRUS) statistical report of central nervous system tumors (2017–2021) evidences an increased prevalence of GBM *IDH*-wt in males (1.6 times more prevalent) than in females, with a median survival duration of 15.0 months in male patients, versus 25.5 months in female patients [51,52]. In addition, experimental investigations have shown that GBM engrafted in animal models displays a reduced growth rate in female subjects compared with their male counterparts [38]. The exact mechanism underlying this pronounced epidemiology is unclear. It has been hypothesized that gender differences may be attributable to the AR signaling axis: the upregulation of the AR signaling pathway has been identified as a contributing factor to the increased male predominance of GBMs [53]. This mechanism appears to play a critical role in the proliferation, migration, and invasion of high-grade gliomas.

Despite decades of research, the identification of prognostic markers for gliomas is limited, with an even smaller subset that may serve as viable therapeutic targets [15,54,55]. We have performed Kaplan–Meier analyses [22] in patients with different AR expression levels to investigate the potential prognostic value of the AR. In our series, higher AR protein expression is associated with significantly poorer OS in glioma patients (*p* = 0.0022) (Figure 3). This finding is consistent with the prior literature indicating that elevated AR expression levels are correlated with poorer prognosis in both IDH-wildtype GBMs and LGGs [15,56] and that the exposure of GBM cells to androgens increases tumor aggressiveness [56].

In our study, we aimed to establish a correlation between AR expression, as assessed through immunohistochemistry, and the findings obtained from DNA sequencing, DNA methylation analysis, and X-chromosome copy number assessment.

Since mutations of the AR gene are associated with higher malignancy of prostate cancer, we investigated the mutational status of the AR to address any possible interactions also in glioma. Most mutations cause a substitution of one amino acid, thus allowing antiandrogens to act as AR agonists and favor cancer progression. The most common AR mutations and their associated drug resistance are p.F877L for enzalutamide and Apalutamide, p.T878A and p.H875Y for Abiraterone, p.W742C and p.V716T for bicalutamide and flutamide [57]. In our cohort, the mutational analysis of the *AR* gene, from exons 4 to 8, revealed no mutations in any of our samples. 

We also performed a DNA methylation analysis of the *AR* and its co-regulatory genes to investigate whether epigenetic mechanisms might regulate *AR* expression, observing some differences in different classes of gliomas.

DNA methylation is a key epigenetic mechanism involved in gene expression regulation across multiple tumor types. Hypermethylation of CpG promoter regions recruits repressive chromatin modifiers, leading to stable silencing of genes, including tumor suppressors, thereby promoting tumorigenesis. In contrast, hypomethylation can reactivate gene expression, contribute to chromosomal instability, and promote oncogene or transposable element activation, further driving cancer progression [58,59,60,61,62,63].

The level of AR protein expression is determined by the transcriptional activity of the *AR* gene. As a result, the methylation status of *AR* gene regulators and variations in the copy number of the X chromosome could significantly impact AR protein expression. In our series, AR CAG repeats and “Intra CAG-CGG repeats” regions resulted hypermethylated in gliomas with low AR protein levels, as well as in LGGs (Appendix A and Appendix AA–D). This indicates that the hypermethylation of the *AR* gene leads to transcriptional repression, resulting in low levels of AR expression, a phenomenon that is primarily evident in LGGs. Moreover, the methylation pattern of AR gene regulators, belonging to the *MAGE* family, revealed noteworthy insights. *MAGEA11* and *MAGEC1* were hypermethylated in AR-negative gliomas (Kruskal–Wallis < 0.05) as well as in *IDH*-mutant low-grade tumors. This suggests that the hypermethylation state of these genes may result in *AR* gene silencing and decreased AR protein expression, and this is primarily evident in *IDH*-mutant lower-grade tumors (Appendix A).

Furthermore, in our series, the X-chromosome polysomy (that is the presence of XX chromosomes in male samples and the presence of more than two X chromosomes in female samples) appeared more frequently in men (*p* < 0.001) and in GBM *IDH*-wt (*p* = 0.05) (Figure 4). X-chromosome polysomy resulted more frequently in MGMT-unmethylated tumors (*p* = 0.037). In MGMT-unmethylated patients, we also observed a trend of higher frequency of *AR* amplification (*p* = 0.06) (Figure 5).

X-chromosome polysomy can play a role in the AR gene copy number and, consequently, in AR protein expression in glioma cells, controlling neoplastic transformation, especially in GBM *IDH*-wt. It is conceivable that in high-grade glioma, the neoplastic cells acquire additional copies of the X chromosome with consequent *AR* gene amplification and AR protein overexpression.

Given their ability to cross the blood–brain barrier, AR antagonists may represent a promising therapeutic avenue for the treatment of GBMs. Currently, a variety of androgen antagonists are under investigation for their efficacy in AR-positive gliomas, particularly in IDH-wildtype GBMs [8,64]. Both enzalutamide and bicalutamide, which are approved for prostate cancer treatment, have shown concentration-dependent antitumor effects in A172, U87MG, and T98G GBM cell lines, with enzalutamide exhibiting a more pronounced impact [38].

Research has demonstrated that enzalutamide significantly decreases the growth of GBM cells in both laboratory and live animal studies. Furthermore, it has been demonstrated to diminish cancer stem cell populations, leading to a 50% improvement in survival rates in an orthotopic patient-derived xenograft model of GBM [38] and to induce apoptosis in GBM cells through the activation of a caspase cascade pathway [47]. The 5α-reductase inhibitor dutasteride, in combination with AR antagonists such as cyproterone or flutamide, exhibits significant antitumor effects in U87 GBM cell cultures [65]. The dutasteride–flutamide combination, notably, showed the strongest inhibition of tumor cell metabolism and proliferation. These findings support the potential of dual AR pathway inhibition as a promising therapeutic strategy for GBMs [65].

An increasing body of evidence indicates that the Epidermal Growth Factor Receptor (EGFR) pathway plays a role in the activation of the AR within GBM cells, as AR activation may also occur via ligand-independent signaling through the EGFR [14]. The pairing of enzalutamide with an EGFR-targeted agent, such as afatinib, may enhance antitumor efficacy in glioma cells by simultaneously targeting both signaling pathways. Afatinib has been used in conjunction with the AR antagonist enzalutamide in IDH-wildtype GBM cells, producing significant outcomes and underscoring the synergistic effects of this drug combination [8,14].

In addition to AR inhibitors, combination therapies incorporating AR-targeted agents alongside other treatment modalities, such as chemotherapy and radiation therapy, are presently under investigation [66]. The objective of these combination strategies is to enhance the effectiveness of AR inhibition and to surmount potential resistance mechanisms. The preliminary results from ongoing trials have demonstrated encouraging outcomes, including better overall survival (OS) rates [8].

The data of the present study suggest that gender differences in glioma patients are significant and could be attributed to the role of AR expression, which is controlled by DNA methylation and X-chromosome aneuploidy. AR nuclear expression, in addition, seems to have a close correlation with glioma grading, being more expressed in GBM *IDH*-wt, and appears to influence the prognosis of gliomas, reducing survival. The present results have some limitations, including the low case number and the retrospective nature of the study. Our findings do not reach statistical significance, likely due to the small sample size, and should be considered trends that require validation in larger cohorts.

## 5. Conclusions

In conclusion, in the present study, we found that the AR seems to be overexpressed in IDH-wildtype GBMs as well as in MGMT-unmethylated tumors and appears to be associated with worse survival. The AR could likely play a role in influencing the biological aggressiveness and the prognosis of adult diffuse gliomas. These points were investigated through immunohistochemical staining, which found further validation in DNA methylation and X-chromosome copy number analyses.

The AR is an ideal therapeutic target in oncology, as androgen deprivation therapy is well tolerated and cost-effective. The results shown here can be potentially interesting as the AR could be a therapeutically useful target for glioma treatment. Moreover, immunohistochemical analysis for the detection of AR positivity is an inexpensive and simple investigation, which can be easily performed in glioma patients.

However, more data are needed to confirm the therapeutic impact of this approach and androgen deprivation therapy warrants further investigation in clinical trials for adult diffuse gliomas. Further research is required to explore and address the resistance mechanisms, the heterogeneity of gliomas, and the potential off-target effects of AR inhibitors. Overall, advancements in targeted AR therapy for adult-type diffuse gliomas offer promising prospects for the development of new treatment approaches.

## Figures and Tables

**Figure 1 biomedicines-13-02379-f001:**
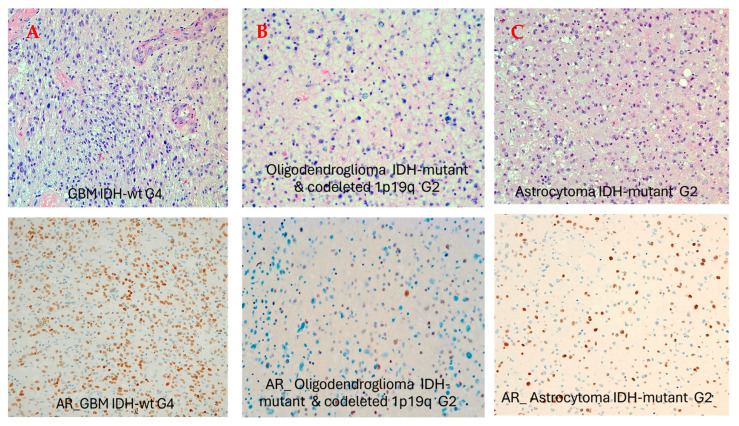
AR nuclear expression in gliomas according to histology and grading. (**A**) a case of AR-positive GBM *IDH*-wt (AR > 70%); (**B**) a case of grade 2 oligodendroglioma *IDH*-mutant with negative AR expression; (**C**) a case of grade 2 astrocytoma *IDH*-mutant with low AR positivity (focal positivity).

**Figure 2 biomedicines-13-02379-f002:**
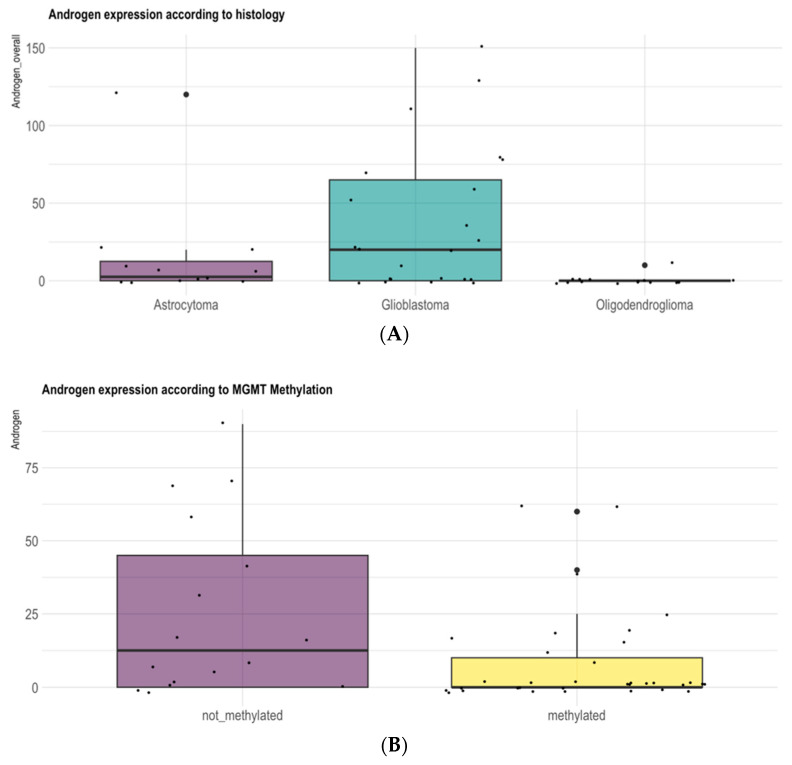
(**A**,**B**) AR expression according to histology and to *MGMT*-methylation status. AR expression was more frequent in GBM *IDH*-wt and in *MGMT*-unmethylated gliomas compared with *MGMT*-methylated tumors (*p* = 0.02).

**Figure 3 biomedicines-13-02379-f003:**
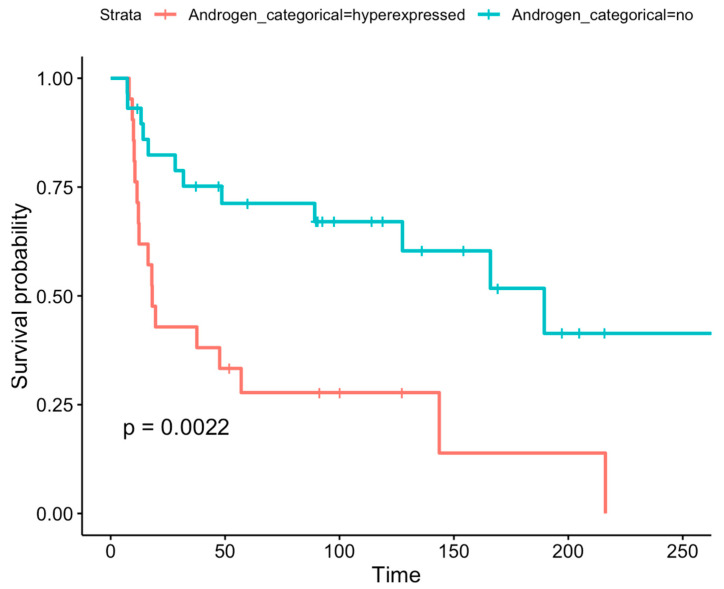
AR expression and OS. In our series AR-positive gliomas exhibit a lower OS than AR-negative gliomas (*p* = 0.0022).

**Figure 4 biomedicines-13-02379-f004:**
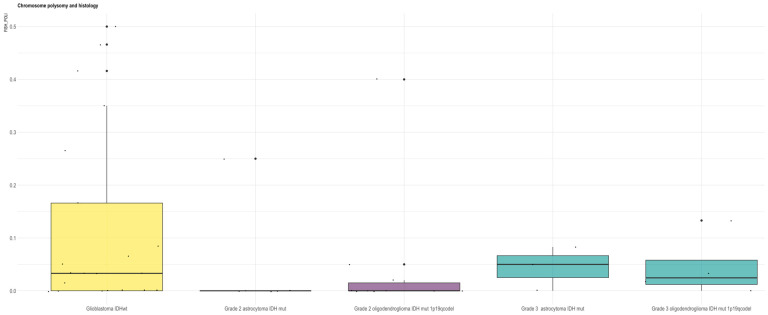
X-chromosome copy number and histology. In our series X-chromosome polysomy resulted more frequently in GBM *IDH*-wt (*p* = 0.05).

**Figure 5 biomedicines-13-02379-f005:**
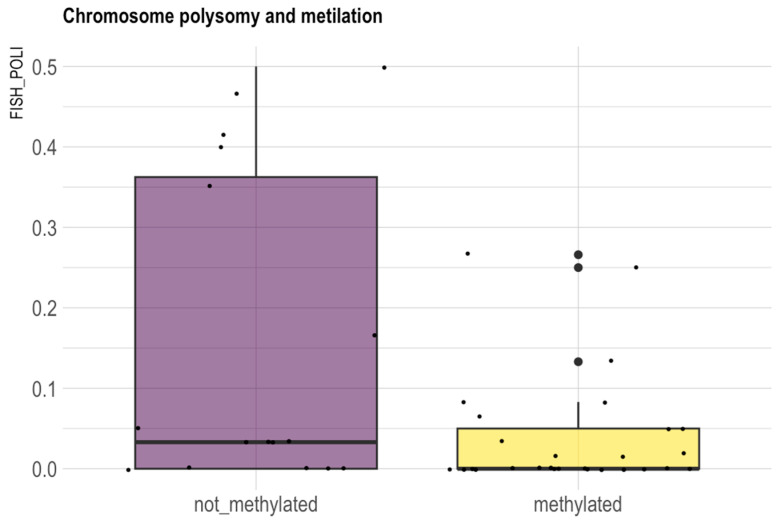
X-chromosome copy number and *MGMT*-methylation pattern. In our series X-chromosome polysomy resulted more frequently in *MGMT*-unmethylated tumors (*p* = 0.037).

**Table 1 biomedicines-13-02379-t001:** Patients clinical-pathological characteristics.

	AR-Negative (n = 29)	AR-Positive (n = 21)	*p*-Value (Chi-Squared Test with Yates Continuity Correction)
**Gender**			
Male	11 (37.9%)	15 (71.4%)	0.04
Female	18 (62.1%)	6 (28.6%)	
**MGMT**			
Unmethylated	5 (17.2%)	11 (52.4%)	0.02
Methylated	24 (82.8%)	10 (47.6%)	
**IDH1**			
WT	10 (34.4%)	15 (71.4%)	0.018
Mutated	19 (65.5%)	6 (28.6%)	
**Age**			
Mean (SD)	48.0	54.0	0.126
Median (Min, Max)	46.3 (26.3, 69.1)	53.7 (29.3, 76.7)	
**Histology**			
Grade 3 astrocytoma IDH-mut	1 (3.4%)	2 (9.5%)	0.018
Grade 3 oligodendroglioma IDH-mut	4 (13.8%)	0 (0%)	
Grade 2 astrocytoma IDH-mut	5 (17.2%)	4 (19.0%)	
Grade 2 oligodendroglioma IDH-mut	10 (34.5%)	1 (4.8%)	
Glioblastoma IDHwt	9 (31.0%)	14 (66.7%)	0.04

**Table 2 biomedicines-13-02379-t002:** Cox Regression Model adjusted for MGMT status in GBM IDH-wt group.

Variable	HR (95%CI)	*p*-Value
GBM MGMT-methylated vs. unmethylated	0.34 (0.12–0.95)	0.04
GBM AR-positive vs. negative	1.68 (0.67–4.26)	0.27

**Table 3 biomedicines-13-02379-t003:** Cox Regression Model adjusted for MGMT status and gender in GBM IDH-wt group.

Variable	HR (95%CI)	*p*-Value
GBM MGMT-methylated vs. unmethylated	0.33 (0.11–0.93)	0.04
GBM AR-positive vs. negative	2.25 (0.79–6.39)	0.12
Gender female vs. male	1.9 (0.71–5.17)	0.19

## Data Availability

The datasets used and/or analyzed during the current study are available from the corresponding author on reasonable request.

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
