# Peer review of "Gender- and Grade-Dependent Activation of Androgen Receptor Signaling in Adult-Type Diffuse Gliomas: Epigenetic Insights from a Retrospective Cohort Study"

_biomedicines, 2025, doi:10.3390/biomedicines13102379_

Round 1

Reviewer 1 Report

Comments and Suggestions for Authors

Abstract:

  • Abbreviations should be double-checked. For example, IDH, WHO, MGMT...
  • Keywords are not included. Please list the appropriate keywords alphabetically.
  • Update the abstract according to journal format.

Introduction:

  • It is not stated why surgery, radiotherapy, and alkylating agent are not effective.
  • A few sentences after the sentence "……..needed to improve outcomes in subsets of patients [3]" would be more appropriate for flow, rather than directly providing information about AR.
  • Examples of studies on experimental therapy methods for glioma could be provided. For example, the sentence "Although surgery, radiotherapy, and alkylating agent chemotherapy are the mainstays of glioma treatment, there is a lack of effective treatments, and individually tailored strategies are needed to improve outcomes in subsets of patients [3]" could be translated as "Although surgery, radiotherapy, and alkylating agent chemotherapy are the cornerstones of glioma treatment, effective treatments are lacking. Although existing studies have tried different therapeutic methods (doi: 10.23736/s0390-5616.22.05718-6), individually tailored strategies are needed to improve outcomes in subgroups of patients [3]." can be replaced with .

Methods:

  • What do you mean by the institution mentioned in line 96? Please clearly state the institution name.
  • No reference is provided for the statistical method used (Kaplan-Meier).
  • The procedural information provided under "2.3. AR immunohistochemistry" is insufficient. The procedure used should be more clearly stated, and a reference should be added.
  • Information regarding the tissue samples used for the study is not clearly stated. When and by which methods were they obtained? How were they preserved throughout the study?
  • A reference should be provided for the summarized method for FISH.

Results:

  • There are shifts in the data in Table 1 due to the tabular format. This may hinder the clear presentation of the information. It should be reorganized.
  • The figures and captions provided under the heading "3.3. Differential DNA methylation analysis of the AR gene promoter region according to immunohistochemical AR expression" are complex and difficult to understand. Please reorganize.
  • The data and related information should be reviewed and reorganized in a simpler and more understandable format.

Discussion:

  • The sentence "AR is a targetable molecule widely used in the treatment of prostate cancer" in line 477 should be referenced.
  • In the discussion, please refer to the tables and figures provided in the results section. For example, "The mutational analysis of the AR gene, from exons 4 to 8, revealed no mutations in all our samples."

References:

  • There are many studies from very old years. Please revise your work according to current literature.

Author Response

COMMENT

“Abbreviations should be double-checked. For example, IDH, WHO, MGMT...

Keywords are not included. Please list the appropriate keywords alphabetically.

Update the abstract according to journal format”.

AUTHOR RESPONSE

We thank the reviewer for the comment. We have introduced a dedicated section for Key Words and Abbreviations, performed a double-check of the abbreviations throughout the text, and formatted the abstract as required by the journal.

You can find it in yellow in the text, lines 51-60.

COMMENT

“It is not stated why surgery, radiotherapy, and alkylating agent are not effective”.

AUTHOR RESPONSE

We thank the reviewer for the comment. We have clarified the reasons why treatments are often ineffective, primarily the inability to fully resect infiltrative tumors, treatment resistance, difficulties delivering drugs across the blood-brain barrier (BBB), and tumor heterogeneity.

You can find it in yellow in the text, lines 68-72.

COMMENT

A few sentences after the sentence "……..needed to improve outcomes in subsets of patients [3]" would be more appropriate for flow, rather than directly providing information about AR.

AUTHOR RESPONSE

We thank the reviewer for the comment. We have clarified and expanded the discussion of the androgen receptor's role in the Introduction section to improve readability and flow.

You can find it in yellow in the text, lines 75-83.

COMMENT

Examples of studies on experimental therapy methods for glioma could be provided. For example, the sentence "Although surgery, radiotherapy, and alkylating agent chemotherapy are the mainstays of glioma treatment, there is a lack of effective treatments, and individually tailored strategies are needed to improve outcomes in subsets of patients [3]" could be translated as "Although surgery, radiotherapy, and alkylating agent chemotherapy are the cornerstones of glioma treatment, effective treatments are lacking. Although existing studies have tried different therapeutic methods (doi: 10.23736/s0390-5616.22.05718-6), individually tailored strategies are needed to improve outcomes in subgroups of patients [3]." can be replaced with.

AUTHOR RESPONSE

We thank the reviewer for the comment. We revised the sentence as suggested by the reviewer and included the reference doi: 10.23736/s0390-5616.22.05718-6 (ref. number 3). You can find it in yellow in the text, lines 75-76.

COMMENT

What do you mean by the institution mentioned in line 96? Please clearly state the institution name.

AUTHOR RESPONSE

We thank the reviewer for the comment. We have mentioned the name of our institution, Bellaria Hospital of Bologna.

You can find it in yellow in the text, line 125.

COMMENT

No reference is provided for the statistical method used (Kaplan-Meier).

AUTHOR RESPONSE

We thank the reviewer for the comment. We have inserted the reference for the statistical method used (Kaplan-Meier) in the text, reference number 22.

COMMENT

The procedural information provided under "2.3. AR immunohistochemistry" is insufficient. The procedure used should be more clearly stated, and a reference should be added.

AUTHOR RESPONSE

We thank the reviewer for this comment. As requested, we have revised paragraph 2.3 to provide a more detailed description of the immunohistochemical method employed, and we have added the appropriate references to support the methodology.

COMMENT

Information regarding the tissue samples used for the study is not clearly stated. When and by which methods were they obtained? How were they preserved throughout the study?

AUTHOR RESPONSE

We thank the reviewer for this comment. We analyzed 50 tissue samples obtained from patients who underwent surgery at Bellaria Hospital in Bologna, between 1999 and 2021. All histological slides/formalin-fixed paraffin-embedded tissue tumor (FFPE) blocks were consistently preserved at the pathology department of our institution. All cases were independently re-evaluated, classified, and graded by a neuropathologist according to currently available criteria (CNS WHO 2021).

You can find it in yellow in the text, lines 124-126.

COMMENT

A reference should be provided for the summarized method for FISH.

AUTHOR RESPONSE

We thank the reviewer for this comment. We have provided references for the methods adopted for FISH (references numbers 17 and 28).

COMMENT

There are shifts in the data in Table 1 due to the tabular format. This may hinder the clear presentation of the information. It should be reorganized.

AUTHOR RESPONSE

We thank the reviewer for the comment. We have modified and reorganized Table 1 format as suggested. You can find it in yellow in the text.

COMMENT

The figures and captions provided under the heading "3.3. Differential DNA methylation analysis of the AR gene promoter region according to immunohistochemical AR expression" are complex and difficult to understand. Please reorganize. The data and related information should be reviewed and reorganized in a simpler and more understandable format.

AUTHOR RESPONSE

We thank the reviewer for the comment. We added the Supplementary figures 1 A,B,C,D to describe in detail the 4 different regions investigated, and we have revised the figure captions to make them clearer and more understandable. In particular, we evaluated 4 different regions of AR gene body: AR CAG repeats (promoter) from coordinates 67545204 to 67545299; the AR region within CAG and CGG repeats (intra CAG-CGG) from 67545904 to 67546136; the AR region before CGG repeat from 67546381 to 67546405; and, finally, AR region within the CGG repeats from 67546492 to 67546664.

AR was found hypomethylated, and therefore probably transcriptionally active, in cases with immunohistochemical positivity of AR (Kruskal-Wallis <0.05). In males, AR CAG repeats and AR INTRA CAG-CGG repeats resulted hypomethylated in AR-positive gliomas. In females, we observed the same trend; however, not statistically significant.

You can find it in yellow in the text, lines 301-313.

COMMENT

The sentence "AR is a targetable molecule widely used in the treatment of prostate cancer" in line 477 should be referenced.

AUTHOR RESPONSE

We thank the reviewer for the comment. We have referenced the sentence "AR is a targetable molecule widely used in the treatment of prostate cancer", as suggested. You can find it in yellow in the text (references 31-36).

COMMENT

In the discussion, please refer to the tables and figures provided in the results section. For example, "The mutational analysis of the AR gene, from exons 4 to 8, revealed no mutations in all our samples."

AUTHOR RESPONSE

We thank the reviewer for the comment. As suggested, we have included references to the tables and figures in the Discussion section. These additions are highlighted in yellow in the text.

COMMENT

There are many studies from very old years. Please revise your work according to current literature.

AUTHOR RESPONSE

We thank the reviewer for this insightful observation. We have revised our work according to current literature, incorporating in the text more up‑to‑date references (see references 37, 38, 39, 40, 41, 47, 53, 64, 66).

Reviewer 2 Report

Comments and Suggestions for Authors

This is a wonderful retrospective study that investigates the expression and epigenetic regulation of the androgen receptor (AR) in a cohort of 50 adult-type diffuse glioma patients. Using immunohistochemistry, DNA methylation analysis, and FISH, the authors demonstrate that AR expression is significantly higher in males, in high-grade gliomas (particularly IDH-wildtype glioblastoma), and in tumors with an unmethylated MGMT promoter. Higher AR expression is also correlated with poorer overall survival. The study provides novel evidence that AR expression is regulated epigenetically through hypomethylation of the AR gene promoter and is associated with X chromosome polysomy in high-grade tumors, suggesting AR is a promising prognostic marker and therapeutic target.

General concept comments

The manuscript addresses a significant and clinically relevant gap in the understanding of glioma biology, particularly the gender disparity observed in glioblastoma. The study's main strength lies in its comprehensive, multi-modal approach, integrating protein expression with detailed epigenetic analyses (DNA methylation of AR and its co-regulators) and cytogenetic analysis (X chromosome copy number). The findings are logically presented, and the discussion effectively situates the results within the existing literature, highlighting their potential therapeutic implications.

The primary weakness, which the authors rightly acknowledge, is the small and heterogeneous sample size (N=50). This limitation affects the statistical power of the analyses, especially within subgroups. For instance, some survival analyses for specific histological subtypes are based on very few patients, making the results preliminary and requiring cautious interpretation. While the retrospective nature is a limitation, the authors have mitigated this by reclassifying all cases according to the latest 2021 WHO criteria, ensuring diagnostic consistency.

Specific comments

  1. Statistical Analysis and Data Presentation:

    • In Table 1, the p-value for "Histology" is listed as 0.018 . It should be clarified what statistical test was used for this multi-categorical variable (e.g., Chi-squared or Fisher's exact test) and what the overall test signifies regarding the distribution of histological types between AR-positive and AR-negative groups.
    • The grouping of tumors into "low-grade" (Grade 2) and "high-grade" (Grade 3 and 4) for methylation analysis is a simplification. Given the significant biological and prognostic differences between Grade 3 and Grade 4 gliomas, this grouping could mask important variations. A brief justification for this approach or a more granular analysis would strengthen the findings.
  2. Interpretation of Results:

    • The overall survival analysis shows a highly significant association between AR positivity and poorer outcomes (p=0.0022). However, the subgroup analyses (e.g., within GBMs stratified by MGMT status) do not reach statistical significance, likely due to the small sample size. The manuscript should more explicitly state that these subgroup findings are trends that require validation in larger cohorts to avoid over-interpretation by the reader.
  3. Clarity and Reproducibility:

    • The methods section is generally well-detailed and provides sufficient information for reproducibility. The description of the AR scoring system, adapted from breast cancer, is clear and appropriate.

Author Response

COMMENT

In Table 1, the p-value for "Histology" is listed as 0.018. It should be clarified what statistical test was used for this multi-categorical variable (e.g., Chi-squared or Fisher's exact test) and what the overall test signifies regarding the distribution of histological types between AR-positive and AR-negative groups.

AUTHOR RESPONSE

We are grateful to the reviewer for the comments and for the appreciation expressed. We used the Chi-Squared Test with Yates’ Continuity Correction. The test means that high-grade tumors, primarily IDH wild-type glioblastomas but also grade 3 gliomas, tend to exhibit higher androgen receptor expression. Due to the small size of our case series, we are only able to demonstrate a trend.

COMMENT

The grouping of tumors into "low-grade" (Grade 2) and "high-grade" (Grade 3 and 4) for methylation analysis is a simplification. Given the significant biological and prognostic differences between Grade 3 and Grade 4 gliomas, this grouping could mask important variations. A brief justification for this approach or a more granular analysis would strengthen the findings.

AUTHOR RESPONSE

We thank the reviewer for these valuable comments. We fully acknowledge the reviewer's comment regarding the distinct prognostic implications of grade 3 and grade 4 tumors. The simplification adopted in our analysis was not intended to suggest prognostic equivalence, but rather to highlight a biological trend—specifically, that androgen receptor expression appears to increase with tumor grade, with higher-grade tumors showing greater expression levels. Due to the small numbers in our case series, we are mainly able to demonstrate a trend in this respect. We are planning a further study, where we intend to implement our case study to give greater statistical strength to the results.

COMMENT

The overall survival analysis shows a highly significant association between AR positivity and poorer outcomes (p=0.0022). However, the subgroup analyses (e.g., within GBMs stratified by MGMT status) do not reach statistical significance, likely due to the small sample size. The manuscript should more explicitly state that these subgroup findings are trends that require validation in larger cohorts to avoid over-interpretation by the reader.

AUTHOR RESPONSE

We thank the reviewer for the comment. We agree that the limitation of this work is represented by a too small cohort.

We have clarified in the text that the present results have some limitations, including the low case number and the retrospective nature of the study. Our findings do not reach statistical significance, likely due to the small sample size, and should be considered trends that require validation in larger cohorts.

You can find it in yellow, lines 676-679.

We thank the reviewer for the suggestion, which is in line with the further study we are planning, where we intend to implement our case study to give greater statistical strength to the results.

Reviewer 3 Report

Comments and Suggestions for Authors

In this work, Gatto et al. investigated androgen receptor expression and epigenetic regulation in adult-type diffuse gliomas. The study was well-conducted and provides interesting insights into androgen receptor signaling in gliomas. However, it contains some limitations, which can nevertheless be addressed through a revision. I therefore recommend a major revision. My specific comments are as follows:

1. The sample size (n=50) is quite small. Splitting it into different subgroups (e.g. AR negative, AR positive, male, female, various histological types) made it worse. For example, there is only a single AR+ oligodendroglioma patient, and reporting OS for one single patient cannot yield a meaningful median OS and the results could be very misleading. Perhaps the authors need to perform a post-hoc power analysis to find out whether the study is adequately powered.

2. For IHC, the authors mentioned that they used an AR scoring system adapted from breast cancer. Was this validated in gliomas? If not, the IRS score threshold may not be appropriate for brain tumors and there could be risks of misclassification of AR positivity. The authors need to provide a strong justification for this. 

3. Details on IHC protocol (the antibody clone, dilution, etc) should be fully described. 

4. The authors stated that mutational analysis was extended to exons 4, 5, and 8 of the AR gene because these exons are associated with drug resistance in prostate cancer. Why prostate cancer? The therapeutic pressure that drives AR mutations in prostate cancer (i.e. anti-androgen therapy) does not exist in gliomas. Thus the biological rationale for analyzing these exons is weak in my opinion. Perhaps the authors need to justify this further.

5. Survival analysis was performed using a univariate Kaplan-Meier plot. However, a number of variables can potentially confound the results (IDH status, grade, MGMT status, etc). It would be preferred to run a multivariate Cox regression model that includes established prognostic variables and clarify which associations hold after adjustment. Hazard ratios and confidence intervals should be reported as well.

6. In many instances, the manuscript presents one sentence per paragraph, which hinders readability. In fact, the flow of these sentences is already very good, so they can be (and should be) combined to form a proper paragraph. In addition, in some sections, bullet points were used. It is preferable to present this content in complete sentences in a scientific paper.

7. A number of supplementary figures appeared in the main text. They should be moved into a separate supplementary file.

Author Response

COMMENT

The sample size (n=50) is quite small. Splitting it into different subgroups (e.g. AR negative, AR positive, male, female, various histological types) made it worse. For example, there is only a single AR+ oligodendroglioma patient, and reporting OS for one single patient cannot yield a meaningful median OS and the results could be very misleading. Perhaps the authors need to perform a post-hoc power analysis to find out whether the study is adequately powered.

AUTHOR RESPONSE

We thank the reviewer for these valuable comments. We agree that the limitation of this work is represented by a too small cohort. As highlighted in the manuscript, our findings do not reach statistical significance and should be considered trends that require validation in larger cohorts.

Due to the limited sample size, a post-hoc analysis with sufficient statistical power is not feasible. We are planning a further study, where we intend to implement our case study to give greater statistical strength to the results.

COMMENT

For IHC, the authors mentioned that they used an AR scoring system adapted from breast cancer. Was this validated in gliomas? If not, the IRS score threshold may not be appropriate for brain tumors and there could be risks of misclassification of AR positivity. The authors need to provide a strong justification for this. 

AUTHOR RESPONSE

We thank the reviewer for this comment. We adopted an AR scoring system adapted from astrocytic tumors.  The staining was scored according to the four-point system (score 0–3). The score 3 consists of brown nuclear staining that is easily visible and involves >50% of cells; score 2: focal brawn nuclear staining areas (50% of cells); score 1: focal moderate nuclear staining in <50% of cells, or pale staining in any proportion of cells not easily seen at low power; score 0: none of the above. The strongest intensity observed in the non-tumoral glial tissue was taken as reference. In tumors with heterogeneous results, the most prevalent score was considered. The scoring was established by a neuropathologist (AS), who was blinded to the clinical data. Tumor samples were classified into AR-positive (AR expression score 1, 2, and 3) and AR-negative (AR score 0).

You can find it in yellow in the text, lines 160-176.

COMMENT

Details on IHC protocol (the antibody clone, dilution, etc) should be fully described. 

AUTHOR RESPONSE

We thank the reviewer for this comment. Immunohistochemistry was performed on an automated stain (Ventana BenchMark, Ventana Medical Systems Inc., Tucson, AZ, USA) applying a pre-diluted monoclonal anti-androgen receptor (Cell Marque, clone SP 107).

You can find it in the text in yellow, lines 160-176.

COMMENT

“Mutational analysis was extended to exons 4, 5, and 8 of the AR gene because these exons are associated with drug resistance in prostate cancer. Why prostate cancer? The therapeutic pressure that drives AR mutations in prostate cancer (i.e. anti-androgen therapy) does not exist in gliomas. Thus the biological rationale for analyzing these exons is weak in my opinion. Perhaps the authors need to justify this further”.

AUTHOR RESPONSE

We thank the referee for this suggestion: the text has been modified as follows in the “Discussion” session:

Since mutations of the AR gene are associated with higher malignancy of prostate cancer, we investigated the mutational status of AR to address any possible interactions also in glioma. Most mutations cause a substitution of one amino acid, thus allowing antiandrogens to act as AR agonists and favor cancer progression. The most common AR mutations and their associated drug resistance are p.F877L for Enzalutamide and Apalutamide, p.T878A and p.H875Y for Abiraterone, p.W742C and p.V716T for Bicalutamide and Flutamide. In our cohort, the mutational analysis of the AR gene, from exons 4 to 8, revealed no mutations in all our samples.

You can find it in the text in yellow, lines 592-599.

COMMENT

Survival analysis was performed using a univariate Kaplan-Meier plot. However, a number of variables can potentially confound the results (IDH status, grade, MGMT status, etc). It would be preferred to run a multivariate Cox regression model that includes established prognostic variables and clarify which associations hold after adjustment. Hazard ratios and confidence intervals should be reported as well.

AUTHOR RESPONSE

We thank the reviewer for the comment. This is a retrospective, hypothesis-generating study. Due to the limited sample size and the low number of events, performing a reliable multivariate analysis (such as Cox regression) with sufficient statistical power is not feasible. As highlighted in the manuscript, the primary limitations of our study lie in the small number of cases and its retrospective design. Our findings do not reach statistical significance, likely due to the small sample size, and should be considered trends that require validation in larger cohorts.

COMMENT

In many instances, the manuscript presents one sentence per paragraph, which hinders readability. In fact, the flow of these sentences is already very good, so they can be (and should be) combined to form a proper paragraph. In addition, in some sections, bullet points were used. It is preferable to present this content in complete sentences in a scientific paper.

AUTHOR RESPONSE

We thank the reviewer for the comment. We have combined sentences to form proper paragraphs and we have removed the bullet points, as suggested.

You can find it in yellow, lines 258-313.

COMMENT

A number of supplementary figures appeared in the main text. They should be moved into a separate supplementary file.

AUTHOR RESPONSE

We thank the reviewer for the comment. We agree that supplementary figures and tables should be moved into a separate supplementary file. We have created a file entitled Supplementary Figures and Tables, which include Supplementary table 1; Supplementary figure 1 A,B,C,D; Supplementary figure 2 A,B,C,D,E,F; Supplementary figure 3; Supplementary figure 4 A, B,C,D; Supplementary figure 5 A, B,C, D, E, F; Supplementary figure 6 A,B; Supplementary figure 7 A,B,C,D; Supplementary figure 8 A, B, C, D, E, F.

Round 2

Reviewer 1 Report

Comments and Suggestions for Authors

Please recheck your plagiarism rate. Reduce the 36% to an acceptable level (e.g., 20%- 25%).

Please move the text Table 1 to the top of the table.

Author Response

COMMENT

Please recheck your plagiarism rate. Reduce the 36% to an acceptable level (e.g., 20%- 25%).

AUTHOR RESPONSE

We thank the reviewer for the comment. We have checked for plagiarism and modified the sections that were similar to other articles. These changes are highlighted in green in the text, lines 109-112; 179-194; 136-266; 682-688; 691-694; 724-726; 730-732; 738-741; 746-759; 784-791.

Furthermore, we used the TextGuard tool to check for plagiarism, and the paper was found to be 80-90% original.

COMMENT

Please move the text Table 1 to the top of the table.

AUTHOR RESPONSE

We thank the reviewer for the comment. We have moved the text of Table 1 to the top of the table, as suggested. You can find it in green, line 296.

Reviewer 3 Report

Comments and Suggestions for Authors

The authors have attempted to revise their manuscript based on my comments. However, there are two points that I am not fully satisfied with. The authors should address these before the paper can be accepted for publication.

1. If a post-hoc analysis is not feasible, the paper would benefit from more comprehensive statistical reporting. For example, the authors could report hazard ratios with 95% CIs from survival models, as well as 95% CIs for subgroup comparisons. In some places, the authors have already reported 95% CIs for medians, but more complete CI reporting (HRs, ORs) is still missing and would strengthen the paper.

2. If a multivariate Cox regression is not feasible, a Cox model adjusting for the strongest prognostic variables should be performed to clarify whether the AR association is independent. For example, in the GBM IDH-wt group, a Cox model adjusting for MGMT status (and, if events permit, sex) would be ideal.

Author Response

COMMENT

If a post-hoc analysis is not feasible, the paper would benefit from more comprehensive statistical reporting. For example, the authors could report hazard ratios with 95% CIs from survival models, as well as 95% CIs for subgroup comparisons. In some places, the authors have already reported 95% CIs for medians, but more complete CI reporting (HRs, ORs) is still missing and would strengthen the paper.

AUTHOR RESPONSE

We thank the reviewer for the comment. We have performed a more comprehensive statistical reporting, including report hazard ratios with 95% CIs from survival models. You can find it in green, section 3.2; lines 325-350.

COMMENT

If a multivariate Cox regression is not feasible, a Cox model adjusting for the strongest prognostic variables should be performed to clarify whether the AR association is independent. For example, in the GBM IDH-wt group, a Cox model adjusting for MGMT status (and, if events permit, sex) would be ideal.

AUTHOR RESPONSE

We thank the reviewer for the comment. We performed a Cox Regression Model for investigating the prognostic impact of AR expression adjusted for MGMT methylation status in the GBM IDH-wt group (Table 2). In this analysis, the MGMT methylation status confirmed its prognostic role (p-value 0.04; HR=0.34 (0.12-0.95)), while AR expression did not reach statistical significance (p-value 0.27; HR=1.68 (0.67-4.26)), likely due to the small sample size. After adjusting the Cox Regression Model for MGMT methylation status and gender, AR expression only showed a trend in impacting survival, without reaching statistical significance (p-value 0.12; HR=2.25 (0.79-6.39)) (Table 3). You can find it in the text in green, lines 343-361.